# NORMSAGE: Multi-Lingual Multi-Cultural Norm Discovery from Conversations On-the-Fly

♠Yi R. Fung, ◇Tuhin Charkaborty, ♠Hao Guo,
♡Owen Rambow, ◇Smaranda Muresan, ♠Heng Ji
♠University of Illinois Urbana-Champaign
◇Columbia University, ♡Stony Brook University
{yifung2,hengji}@illinois.edu

## Abstract

Knowledge of norms is needed to understand and reason about acceptable behavior in human communication and interactions across sociocultural scenarios. Most computational research on norms has focused on a single culture, and manually built datasets, from non-conversational settings. We address these limitations by proposing a new framework, NORMSAGE[1], to automatically extract culture-specific norms from multi-lingual conversations. NORMSAGE uses GPT-3 prompting to 1) extract *candidate norms* directly from conversations and 2) provide *explainable self-verification* to ensure correctness and relevance. Comprehensive empirical results show the promise of our approach to extract high-quality culture-aware norms from multi-lingual conversations (English and Chinese), across several quality metrics. Further, our relevance verification can be extended to assess the adherence and violation of *any* norm with respect to a conversation on-the-fly, along with textual explanation. NORMSAGE achieves an AUC of 94.6% in this grounding setup, with generated explanations matching human-written quality.

## 1 Introduction

Norms are rules that embody the shared standards of behaviors amongst cultural groups and societies (Schwartz et al., 2012). These include *social conventions* (*e.g.,* it's good to shake hand with your opponent even if you lost); *behavior guidances* (*e.g.,* it's wrong to hurt a pet); and *general concepts* (*e.g.,* it's nice to be smart) (Forbes et al., 2020; Ziems et al., 2022). Knowledge of norms in general, and sociocultural norms in particular, is essential if we are to equip AI systems with capability to understand and reason about acceptable behavior in communication and interaction across cultures.

Yet, research on computational modeling of social norms is still in its early stages. SOCIAL-CHEM-101 (Forbes et al., 2020) and the MORAL INTEGRITY CORPUS (Ziems et al., 2022) introduce large crowdsourced datasets of social norms and moral judgements conceptualized as free-text Rules-of-Thumb (RoT) grounded in given situations. While this has led to significant advances in computational modeling of social norms, there are two important limitations. First, while models trained on these grounded RoTs can adapt to new situations, they often struggle to uncover new norms and to model richer context. Second, the norms covered in these studies are primarily Reddit-based and US-centric (Bianchi, 2022).

In this paper, we propose to address current shortcomings with the following contributions:

- **A new task: conversation-based multilingual and multi-cultural norm discovery (Sec 2).** Norms manifest themselves in day-to-day conversations, either by being implicitly adhered to or violated. For example, we observe in Fig 1 input a conversation between a Taiwanese-American parent and son about *"not wasting food"*. We define a new task aimed at extracting norms from conversation context, beyond single statements and question-answer pairs from recollected summaries. To support this task, we also introduce a collection of conversations in English and Chinese spanning multiple genres (e.g., TV shows, videos, text messages), topics, and cultures.

- **NORMSAGE, a zero-shot language model prompting and self-verification framework for norm discovery from conversations (Sec 3).** First, we extract candidate norms using GPT-3 (Brown et al., 2020) prompting that takes the conversational and social context into account (e.g., culture, social relations, discussion topic, etc.), as shown in the Fig 1 *dvr(·)* operation for initial norm discovery. Second, to ensure norm correctness and relevance, we propose the idea

---

[1]Our code and data are available at the Github repo here.

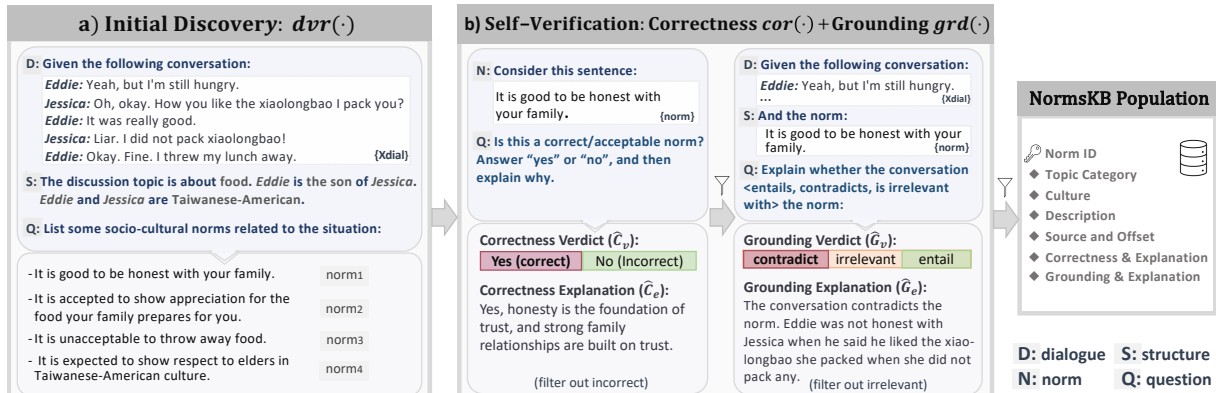

Figure 1: The key idea of **NORMSAGE** is prompting & verification for conversation-grounded norm discovery. Norm candidates are first discovered from dialogues, through the ***dvr(·)*** operation. They are then checked for correctness and relevance, through ***cor(·)*** and ***grd(·)***, before populating into the knowledge base of norms, **NormsKB**.

of explanation-aware self-verification through prompting-based operations ***cor(·)*** and ***grd(·)***. As shown in Fig 1(b), our formulation here involves a question assessing the correctness of a candidate norm, and another probing how the norm grounds to its source dialogue. The entailment reasoning in ***grd(·)*** can be further used to assess the adherence or violation of *any* norm in a conversation, with textual explanation.

- **Comprehensive evaluation setup (Sec 4).** We assess the quality of culture-aware norm extractions along several dimensions. Our method discovers significantly more relevant and insightful norms for conversations on-the-fly (selected as *"best"* ≥70% cases by human judgement) compared to baselines developed over Reddit-based rules-of-thumb annotations (Forbes et al., 2020; Ziems et al., 2022). Additionally, the norms discovered from Chinese conversation exhibit limited performance difference compared to the norms discovered from English conversation in terms of relevance and correctness ($\Delta \leq 5\%$). The culture-specific norms are also promising in quality, with humans being able to identify the culture from these norms at 80% accuracy.

- **NormKB**, a knowledge base of over 20k unique norms with correctness and grounding explanations, extracted from multi-lingual multi-cultural conversations.

## 2 Task Formulation and Source of Data

### 2.1 Task Formulation

We define the **conversation-based, multi-lingual multi-cultural norm discovery** problem as follows. Given a conversation scenario ($\mathbf{X}_{\text{dial}}$) in one

of the pre-defined target languages (*e.g.*, English, Chinese), we aim to derive a set of relevant norms $\mathbf{N} = \{\mathbf{n}_1...\mathbf{n}_m\}$ in English, for a unified representation. If background information ($\mathbf{X}_b$) about the conversation context or speaker profile is available, such as from the Wikipedia summary of a TV show, it can optionally be incorporated to enrich norm discovery. Because handling hallucination and achieving transparency are important for norm discovery, we introduce checking **norm correctness** and **grounding** as supplementary **verification** tasks. We aim to filter out incorrect norms, by deriving a correctness verdict $\mathbf{C}_v \in \{\mathbf{1} : yes, \mathbf{-1} : no\}$, along with a confidence probability and natural language explanation. Additionally, we aim to filter out non-insightful norms, by deriving a grounding inference $\mathbf{G}_v \in \{\mathbf{1} : entail, \mathbf{0} : irrelevant, \mathbf{-1} : contradict\}$, along with confidence probability and explanation.

Our proposed task is innovative in several key aspects. It is the first task to define automatically discovering norms from dialogue data, which best reflects human communication on-the-fly. In addition, it is the first task on discovering multicultural norms from multilingual sources, which is then used to construct a **NormsKB** that can better represent diverse socioethnic or demographic groups. Finally, grounding the discovered norm with dialogue examples, confidence score, and natural language explanations enables our norm discovery approach to be explainable and self-supervised. This verification process can be used to check whether a conversation adheres or violates any given norm.

### 2.2 Source of Data

In collecting source data for norm discovery from

| | Source of Data | Lang | Topic | # Tok | # Trn |
|---|---|---|---|---|---|
| **Single Culture** | Big Bang Theory (BBT) | EN | the life of nerdy, brilliant scientists in California | 29,682 | 3,468 |
| | Friends (F) | EN | Manhattan adult life, friendship, dating | 26,197 | 2,849 |
| | How I Met Your Mother (HIMYM) | EN | romance centered around NYC | 29,423 | 3,785 |
| | Grey's Anatomy (GA) | EN | medical environments centered in Seattle | 23,341 | 3,117 |
| | Castle (C) | EN | law and justice centered around NY police | 38,880 | 4,142 |
| **Cross-Culture** | Fresh off the Boat (FOB) | EN | Taiwanese family in America | 26,056 | 4,129 |
| | Never Have I Ever (NHIE) | EN | Indian girl coming-of-age in the US | 39,847 | 5,637 |
| | Blackish (B) | EN | African American family in the suburbs | 33,993 | 5,103 |
| | Citizen Khan (CK) | EN | Pakistani family in Britain, patriarchism | 22,985 | 3,198 |
| | Outsourced (O) | EN | American salesman, outsourced to India | 1,464 | 123 |
| **MultiLing** | American Factory (AF) | EN,CN | Chinese company open factory in Ohio | 10,840 | 1,138 |
| | Real-World Negotiations (RWN) | EN,CN | US-China talks, street vendor purchase talks | 19,487 | 1,758 |
| | LDC2022E11 CCU TA1 Dev. | CN | text msg, phone calls, online videos | 1.2M | 102k |
| | Total | - | - | 1.5M | 140k |

Table 1: We list out the sources of our raw data here, along with description of their language (EN - English, CN - Chinese), topic domain, and statistics (# tok - number of spoken tokens, # trn - number of dialogue turns).

conversations, we seek data that involve dialogue exchanges mimicing or reflecting real-world communication. Secondly, the data should ideally span diverse topics and societal or cultural groups. In practice, it is generally difficult to obtain large-scale, in-the-wild data for norm discovery due to privacy concerns, as well as sparsity of interesting human interaction occurrences. TV shows and movies offer an interesting alternative, as they represent naturally occurring conversations often involving multicultural exchanges. We expand on the predominantly single-cultured TVQA dataset (Lei et al., 2018), and collect a set of TV and movies that cover different cultures, all of which are primarily in English. We also include several multilingual (Chinese and English) conversations from real-world chats, negotiations, and documentaries, to explore norm discovery adaptability in diverse data settings, as detailed in Table 1. In particular, the LDC data is a large, compiled release of several thousand SMS/chat, phone conversations, and YouTube/Bilibili video scenarios (LDC, 2022). For each source of data, we collect the corresponding textual summary from Wikipedia, if available. Lastly, we preprocess dialogue transcripts into chunks ($\mathbf{X}_{\text{dial}_{1..N}}$) every $k = 5$ lines as the individual data points in our task consideration.

## 3 NORMSAGE Framework

### 3.1 Initial Norm Discovery

Pretrained language models store implicit knowledge about the world learned from large-scale text collected around the internet (Petroni et al., 2019; Li et al., 2023b). We frame conversation-based norm discovery as a series of natural language prompts, each with a directed question for the pretrained GPT-3 Davinci language model to reason with its internal knowledge and generate an answer response. To **d**isco**ver** an *initial* set of candidate norms from conversation data, we introduce the *dvr*(·) operator, which concatenates **D**, a template header describing the nature of the context data followed by the dialogue input {$\mathbf{X}_{\text{dial}_i}$}, with **Q**, a directed question describing the norm discovery task, as input for the PLM to generate response. We follow the general prompt development guideline of wording **D** and **Q** in clear and concise language.

**Structure Enhancement** A shortcoming observed in standard prompting is that the norms discovered may lack well-formedness and taxonomy for categorizing information specific to different cultures and topics. To encourage greater level of detail and structure in *dvr*(·) outputs, we investigate adding to the prompt input:

**S** – a building block in the text template consisting of either frame**s** defining the expected structure of norms (see Fig 2), or cultural indicator**s** encouraging topic-specific taxonomy (see Fig 1a). These cultural indicators are extracted automatically through prompting on the background summary if the data is available. Further details of this process are included in A.1.

### 3.2 Self-Verification with Correctness Checking & Explainable Grounding

For each discovered norm, we add a *cor*(·) operator to check the correctness of norms through prompting. This prompting operator follows the

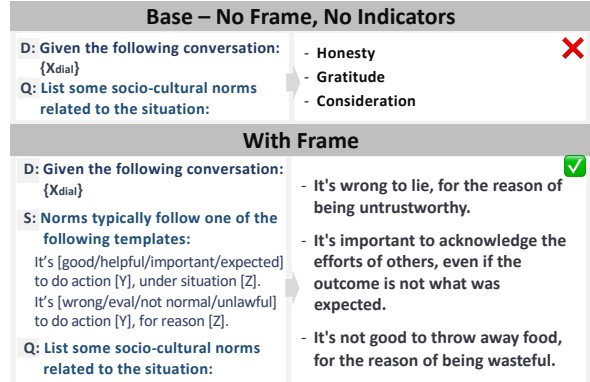

Figure 2: A comparison of ***dvr(·)***–NORMSAGE in *base* form and with *framing* structure.

natural language template of: *"Consider this sentence: {$\underline{n}_i$}. Is this a correct/acceptable social norm? Answer 'yes' or 'no', and then explain why."*. The output from ***cor(·)*** consists of both a correctness verdict, $\mathbf{C_v}$ and a subsequent explanation, $\mathbf{C_e}$, in a single natural language response generation (Fig 1b). We further derive a confidence score ($\hat{\mathbf{C}}_\mathbf{p}$) for the correctness verdict, by normalizing the probability of token generation for $\mathbf{C_v}$ ='yes' with the probability of token generation for $\mathbf{C_v}$ ='no'. Norm candidates with correctness probability below a tunable threshold of $\theta = 0.7$ are filtered out.

As norms are subjective in nature and language models have the risk of hallucination in their output predictions, we further safeguard norm discovery with a ***grd(·)*** operator for determining whether the hypothesized norm discovery can be groundable to its situation premise. We reformulate the explainable NLI setting (Camburu et al., 2018; Huang et al., 2022, 2023) for our new conversation-based norm grounding task, and devise the following natural language template: *"Explain whether the conversation <**entails**, **contradicts**, or **is irrelevant with**> the given norm"*. The output from ***grd(·)*** consists of the grounding verdict, $\mathbf{G_v}$, along with the explanation, $\mathbf{G_e}$. We get the grounding relevance score by normalizing the token probability being either *"entail"* or *"contradict"* in $\mathbf{G_v}$ over the sum of probability of generating *"entail"*, *"contradict"*, or *"irrelevant"*. Then, we filter out the norm candidates with grounding score below a tunable threshold, $\gamma = 0.6$.

Finally, when populating **NormsKB** with new norm discoveries, we disregard norms that duplicate existing ones in the norms library. We flag norms as duplication if their BERT(n) embeddings

(Devlin et al., 2019) exceed a threshold of cosine similarity with any previously discovered norm. The threshold is empirically set to $\sigma = 0.95$.

## 4 Evaluation and Results

We organize this section as follows. In Sec 4.1, we detail experiments on norm discovery from conversations, including multi-lingual and cross-culture scenarios. Then, in Sec 4.2 and Sec 4.3, we analyze the effectiveness of self-verification mechanisms for norm discovery correctness, and relevance grounding between the norm and conversation, respectively. Finally, in Sec 4.4, we discuss the cost and resource contributions.

### 4.1 Intrinsic Norm Discovery Evaluation

**Baselines** While there has been no prior work on norm discovery based on conversations, we include the following baselines trained on a different data format, as valuable points of comparison. $\mathbf{NMT_{gen}}$ is a GPT2-XL trained on SOCIALCHEM101 (Forbes et al., 2020), while $\mathbf{SOCIALCHEM_{rtv}}$ retrieves the most relevant SOCIALCHEM101 rule-of-thumbs for a dialogue based on their embeddings encoded from pre-trained BERT (Devlin et al., 2019). We also consider $\mathbf{pMT_{gen}}$, which is a generator trained on the MORAL INTEGRITY CORPUS (MIC) (Ziems et al., 2022), and $\mathbf{MIC_{rtv}}$, which retrieves the most relevant MIC rules-of-thumb for a dialogue based on their embeddings encoded from pre-trained BERT. In addition, we include $\mathbf{T0_{pp}}$, a T5 model 16x smaller than GPT-3 that is trained on tasks formulated as natural language prompts (Sanh et al., 2022). In our proposed GPT-3 prompting with self-verification approach, we consider the variant that utilizes structural enhancement in initial norm discovery as $\mathbf{NORMSAGE^+}$. We also include **GPT-3** prompting, without self-verification and without structure enhancement, as an additional baseline. Moreover, we include an open and accessible alternative, $\mathbf{NORMSAGE_{mini}}$, which consists of a GPT2-XL finetuned on the norm discovery outputs from **NORMSAGE**.

**Metrics** We proposed to evaluate norm discovery from conversations along the dimensions of relevance, well-formedness, correctness, insightfulness, and relatableness, using a Likert scale of 1 ("awful") to 5 ("excellent"). Definitions and assessment guidelines of these metric categories are included in A.2.1. We also assess what percentage of norms discovered from the various methods are

| | Relevance | Well-Formedness | Correctness | Insightfulness | Relatableness | Best |
|---|---|---|---|---|---|---|
| SOCIALCHEM$_{rtv}$ | 3.8 | 4.0 | 3.9 | 3.8 | 3.9 | 6.2% |
| NMT$_{gen}$ | 3.1 | 3.9 | 3.5 | 3.4 | 3.7 | 0.0% |
| MIC$_{rtv}$ | 3.3 | 3.1 | 3.6 | 3.4 | 3.2 | 7.1% |
| PMT$_{gen}$ | 2.2 | 3.2 | 3.1 | 3.0 | 2.5 | 0.0% |
| T0$_{pp}$ | 2.7 | 2.0 | 2.0 | 2.1 | 2.1 | 0.0% |
| NORMSAGE | 3.9 | 2.8 | 4.1 | 3.5 | 3.6 | 51.3% |
| NORMSAGE$^+$ | **4.5** | **4.5** | **4.6** | **4.2** | **4.7** | **70.4%** |
| - GPT-3 | 3.0 | 2.8 | 2.8 | 2.8 | 3.6 | 25.5% |
| - NORMSAGE$_{mini}$ | 3.8 | 4.2 | 3.7 | 3.4 | 4.0 | 11.9% |

Table 2: Likert scale (1-5) ratings across five quality metrics and the percentage of norm discovery outputs considered as **"best"** for a given dialogue scenario, averaged over 100 data samples.

rated as ***best*** overall, for each dialogue scenario. More than one norm can be considered as "best".

**Setting** We crowdsource on Amazon Mechanical Turk for human assessment. Each HIT ("submit" task) consists of a dialogue scenario, an assessment metric, and sets of norms representing the norms discovered from each de-identified generation or retrieval approach. Following crowdsourcing guidelines (Sheehan, 2018), we provide definitions and examples for each assessment metric. Workers undergo a qualification test ensuring a $\geq 95\%$ HIT rate, comprehension of task instructions, and native fluency in the conversation language. For assessing norms in cross-cultural (pairwise) scenarios, we ensure that the workers have basic familiarity with both cultures. Based on these worker selection criteria, we assign each example to ten distinct workers, and reject poor quality hits. Workers take 1-2 minutes per norm comparison task, and are rewarded a $15 hourly rate.

**Results** Table 2 shows our norm discovery intrinsic evaluation results on English conversation scenarios. As we can see, our proposed norm discovery approach, **NORMSAGE**, outperforms baselines across all dimensions when enhanced with either frame or structured indicators. A major limitation of baseline approaches is poor portability to conversation domains. The performance of **SOCIALCHEM$_{rtv}$** and **MIC$_{rtv}$** shows that simply retrieving pre-annotated norms results in the norms being less relevant and insightful for new conversations. Compared to the retrieval baselines, the generation baselines, **NMT$_{gen}$** and **PMT$_{gen}$**, perform even worse. This suggests that the domain gap in situation context between curated Reddit post headers (previous approaches) and in-situ conversations (current task) poses an even greater bottleneck for norm discovery here. **NORMSAGE** overcomes challenges in domain portability through operationalizing zero-shot language model prompting

for conversation reasoning and norm discovery.

#### 4.1.1 Multi-Lingual Norm Discovery

As a case study of the multi-lingual task setting, we investigate norm discovery performance on Chinese conversations, since Chinese is a widely spoken and studied language separate from the western language systems. As visualized in Fig 3, the norms discovered from Chinese conversations are high-quality in detail and correctness.

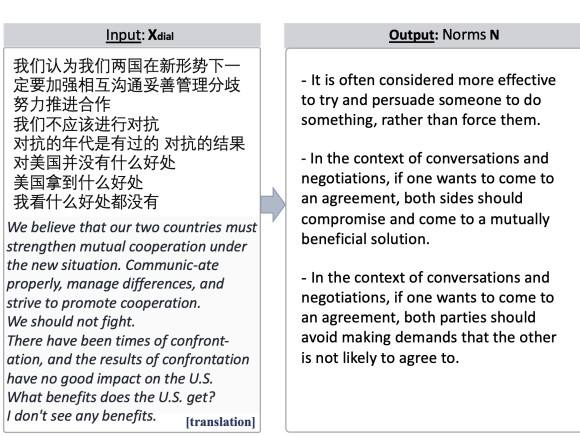

Figure 3: Example illustration of **NORMSAGE** norm discovery results on Chinese conversations, demonstrating our method's multi-lingual capability beyond English data.

We also perform an investigation on the quality of norms discovered from Chinese conversations compared to norms discovered from English conversations. We measure the stand-alone correctness and insightfulness of norms discovered from multi-lingual setting, on a 1-5 Likert scale. The results in Table 4 indicate that NormSage exhibits relative consistency in norm discovery across both English and Chinese conversations, with $\Delta \leq 9.5\%$ observed in terms of relevance and correctness. Interestingly, norms discovered from English data receive slightly higher rating for insightfulness ($\leq 9.5\%$) but lower ratings for relatableness ($\leq 2.1\%$), potentially due to the dialogue

| Culture-Specific Norm Discoveries | Source |
|---|---|
| In **Pakistani** culture, it is not uncommon for the bride and groom to not meet each other until wedding day.[•] | CK |
| In **India**, it is considered polite to always offer food and drink to guests, even if they decline.[◇] | O |
| In **Taiwanese** culture, it is more common to have a heavier lunch, such as rice and vegetables.[○] | FOB |
| In **British** culture, it is normal for the bride and groom to meet each other before wedding.[•] | CK |
| In **American** culture, it is common to have a light lunch, such as a salad or sandwich.[○] | O |
| In **America**, it is more common to just let guests decline if they don't want anything.[◇] | FOB |

Table 3: Visualization of some examples in culture-specific norm discovery. We denote the pairs of contrasting norms across cultures with special symbols ($\diamond, \circ, \bullet$).

nature. The English sources involve movies and shows, which tend to be more creative (insightful) and less formal (biased dialogues may lower norm correctness). Finally, our prompting-based method for norm discovery from conversations can be applied to other languages that the large language model backbone, GPT-3, has been pretrained on.

| | CN Conv. Norm | $\Delta\%$ w. EN Conv. Norm |
|---|---|---|
| **Relevance** | 4.3 | -4.4 |
| **Well-Formedness** | 4.8 | +6.6 |
| **Correctness** | 4.6 | 0.0 |
| **Insightfulness** | 3.8 | -9.5 |
| **Relatableness** | 4.8 | +2.1 |

Table 4: Likert-scale (1–5) rating of norms discovered from Chinese (CN) conversation, and comparison with norms discovered from English (EN) data.

### 4.1.2 Culture-Specific Norm Discovery

To evaluate the correctness of culture-specific norms, we design a pairwise culture comparison setting. Specifically, we run a pretrained **BART-LARGE-MNLI** model (Lewis et al., 2020; Williams et al., 2018) on pairs of norm from different cultures, and randomly select 10 pairs of norm that are determined as "contradiction" with each other, for each of the cross-culture scenarios in our dataset (*e.g, Fresh off the Boat*, *Outsourced*, etc.). Then, we mask the culture identities in the pairs of norm, and ask human annotators familiar with both cultures to identity which culture each norm belongs to from binary options. The results, as shown in

| Culture Comparison | Source | Id | Rating |
|---|---|---|---|
| **American** vs **Taiwanese** | FOB | 74 | 4.2 |
| **American** vs **Indian** | O | 82 | 4.0 |
| **Western** vs **Muslim** | CK | 91 | 4.5 |
| **African American** vs **Caucasian American** | B | 73 | 4.0 |
| Average | - | 80 | 4.1 |

Table 5: We evaluate culture-specific norm discovery via culture **id**entification accuracy (%) between pairs of norms and also Likert scale **rating** (1–5) on the norms.

Table 5, indicate that the culture-specific norms discovered from **NORMSAGE⁺** are promising, with human annotators achieving 80% identification accuracy. The inter-annotator agreement for the culture-identification task, as measured by Fleiss Kappa, is 60.8%, which signals moderately good agreement. Some of the error cases in culture comparison of norm discoveries may be due to subjectivity of the assessment task (for example, norms about whether the Caucasian American or African American cultural group is more likely to discuss controversial topic).

### 4.2 Extrinsic Evaluation on Norm Correctness Self-Verification

**Baselines** We compare *cor*(·)-**NORMSAGE**, the mechanism which checks for correctness in our proposed norm discovery approach, with:

- **BART–SOCIALCHEM–AG**: We finetune a BART-LARGE model (Lewis et al., 2020) to perform regression and predict *anticipated agreement* scores in the SOCIAL-CHEM-101 dataset (Forbes et al., 2020), which reports on a 1–5 scale how much people agree with the rules-of-thumb.
- **T0$_{pp}$**: For prompt template, we follow the same principles as for *cor*(·)-**NORMSAGE** to first introduce the norm and then ask a directed question about its correctness.

For reproducibility in this classification task, we set temperature to 0 during prompting generation.

**Metrics** Norm correctness verification is a two-class classification problem. We measure performance in terms of classification accuracy (**Acc**) and area under the ROC-curve (**AUC**). We also assess the textual explanations of correctness predictions through human rating on a 1–5 Likert scale.

**Data Setting** We randomly select a set of norms discovered from *dvr*(·)-**NORMSAGE**, and ask crowd workers (paid $15/hr) to provide correctness labels and explanations. We keep the data in which the correctness label has been agreed upon by at least

three annotators. In total, we derive a set of 100 norms that are class-balanced with half "correct" and half "incorrect" labels.

**Results** Table 6 shows the results of norm correctness verification. The zero-shot prompting approaches, *cor*(·)-NORMSAGE and $T0_{pp}$, achieve the best performance for this task. Interestingly, we observe that $T0_{pp}$ has a higher Acc and AUC score than *cor*(·)-NORMSAGE. We believe this may be due to the fact that the norms have been generated by the same underlying GPT-3 language model backbone now used to verify the data. As a result, the language model may tend to associate the semantic content or linguistic artifacts in the norm as correct. However, *cor*(·)-NORMSAGE remains a competitive method for verifying norm correctness and has the added advantage of generating high-quality explanations for its predictions, enabling transparent reasoning. In contrast, $T0_{pp}$ prompting cannot generate reasonable explanation on norm correctness beyond circular reasoning. Finally, even though BART-SOCIALCHEM-AG is finetuned on SOCIAL-CHEM-101 *"agreement"* annotations, it does not perform as well in our norm correctness task. This is likely due to the lack of culture-specific rules-of-thumb annotated in the SOCIAL-CHEM-101 dataset to train a robust correctness verification model.

| | Acc | AUC | Expl |
|---|---|---|---|
| BART-SOCIALCHEM−AG | 75 | 78.0 | N/A |
| $T0_{pp}$ | 82 | 92.0 | 1.0 |
| *cor*(·)-NORMSAGE | 81 | 85.3 | 4.2 |

Table 6: Classification results (%) and Likert rating (1–5) on the generated explanations for norm correctness.

### 4.3 Extrinsic Evaluation on Norm Grounding

Norm grounding is utilized in the self-verification process of our norm discovery, as *grd*(·)–NORMSAGE. This subtask extends to identifying norm adherence and violation in conversations, and stands as an important downstream application.

**Baselines** We compare *grd*(·)-NORMSAGE with existing NLI-based approaches, including **BART-MNLI**, a BART-LARGE model (Lewis et al., 2020) pretrained on Multi-genre NLI (Williams et al., 2018); **BART-DIALNLI**, a BART-LARGE pretrained on Dialogue NLI (Welleck et al., 2018); and **T5-eSNLI**, a T5 (Raffel et al., 2020) pretrained on explainable NLI (Camburu et al., 2018). We further consider $T0_{pp}$, introduced in Sec 4.1. For its

prompt template here, we follow the same principles as for *grd*(·)-NORMSAGE to introduce the dialogue context and norm, and question about its adherence/violation relevance. We also evaluate training a BART-LARGE model on the grounding outputs of NORMSAGE, as *grd*(·)-NORMSAGE$_{mini}$. Similar to Sec 4.2, we remove all model randomness.

**Metrics** Similar to the norm correctness task, we evaluate norm grounding in terms of classification accuracy (**Acc**) and area under the ROC-curve (**AUC**). For both metrics, we consider a two-class scenario to determine whether a norm is *relevant* (either entailed or contradicted) or *irrelevant* to a dialogue. We also consider a finer-grained three-class scenario on whether the norm is *entailed*, *contradicted*, or *irrelevant*. As one speaker may violate a norm while another points out adherence issues, we evaluate three-class norm grounding conditioned on speaker localization (*i.e.,* for the dialogue of a given speaker), if speaker identity is available. Finally, we use human assessment to evaluate grounding prediction explanations on a 1–5 Likert scale.

**Data Setting** We randomly sampled dialogue chunks and norm candidates to create a set of 100 data annotations with class-balanced <*contradict, irrelevant,* or *entail*> labels and corresponding explanations. The annotations include speaker identities for dialogues with known speakers. During three-class inference, we consider two data input variants for each method, with and without speaker localization, to investigate whether deeper differentiating of who violates and who adheres to a norm helps. Speaker localization is incorporated into NLI classification models by using only the dialogue lines from the speaker of interest as input. It is incorporated into the prompting-based *grd*(·)-NORMSAGE and $T0_{pp}$ approaches via a question in the prompt template, "Explain whether what's spoken by [speaker ID] <*entails, contradicts, is irrelevant with*> the norm". We report three-class grounding results utilizing the data setting, with or without speaker localization, that maximizes each model's performance and clarity the details next.

**Results** Speaker localization does not benefit NLI classification models for two-class norm grounding, possibly due to a domain shift in semantics after isolating a speaker's dialogue, but it plays a crucial role in prompting-based approaches for three-class norm grounding. In particular, *grd*(·)-NORMSAGE significantly outperforms all other

| Dialogue Situation | Discovered Norms | Grounding Explanation | Label |
|---|---|---|---|
| **Dave**: No. Definitely booked. 
 **Mr. Khan**: What?! Do know who I am? Hello! Mr Khan, community leader! Next President of Sparkhill Pakistani Business Association! 
 **Dave**: I'm sorry 
 **Mr. Khan**: Right, that's it . I want to speak to the proper manager. 
 **Dave**: I am the property manager. | It's important to listen to others and give them a chance to speak. | Mr. Khan is not listening to Dave and he is not giving Dave a chance to speak . | -1 |
| **Beckett**: Sure I can, until a jury tells me otherwise. 
 **Creason**: You are wasting my time. Detective, look, I told you exactly what I was doing last night. 
 **Beckett**: Right. You were at the club. They said that you made quite the entrance [...] | It is generally considered impolite to make lewd comments. | What's spoken by Creason is irrelevant with the norm. | 0 |
| **Jessica**: [...] Well, those kids, they just don't know, that's all. It just – it just take time to get used to something different. 
 **Eddie**: I hate it here! I want to go back to D.C. 
 **Jessica**: Eddie, that's not possible. We are here now. We have to make the best of it . Like I am doing with this neighbor woman. You think I like pretending Samantha isn't carrying a baggie of dog poops in her hand? No! I don't like this ! ...But I am trying! | It is also considered polite to try to make the best of a situation, even if you do not like it | The mother is trying to make the best of the situation even though she does not like it | 1 |

Table 7: Norm grounding example results, randomly sampled for each class from *{-1: Contradict, 0: Irrelevant, 1: Entail}*. We underline the utterance-level provenance of the input, where entailment or contradiction is found.

baselines in norm grounding, as shown in Table 8. For three-class norm grounding, it achieves accuracies of 68% without and 80% with speaker localization. The findings suggest that speaker localization can allow for a fairer evaluation accounting for differences in norm grounding across speakers, and help models reason over the perspective and context of a speaker. We further highlight that *grd(·)*-NORMSAGE generates easy-to-follow and insightful norm grounding explanations that are preferred over human-written ones in around 40% of cases. Examples of the grounding and explanation of norms discovered from conversations are visualized in Table 7. It is worth noting that even though the dialogue inputs may contain improper grammar, involving noisy or informally styled textual dialogue, this is exactly the nature of real-world conversations which makes our novel norm discovery on-the-fly task setting introduced unique and challenging. Despite the noisy nature of dialogue inputs, Table 7 shows that our large language model prompting mechanism can still effectively reason over the <entailment, irrelevance, contradiction> nature of grounding between a given dialogue input and a given norm with proper explanations.

### 4.4 Resource Contribution

We discovered over 20,500 unique norms, of which 1,250 are culture-specific. On average, NORMSAGE discovers norms at a rate of 8.6 sec-

|  | (2-Class) | | (3-Class) | | |
|---|---|---|---|---|---|
|  | ACC | AUC | ACC | AUC | **Expl** |
| **BART-MNLI** | 36 | 31.5 | 32 | 27.3 | N/A |
| **T5-eSNLI** | 50 | 22.9 | 33 | 20.0 | 1.92 |
| **BART-DIALNLI** | 67 | 70.1 | 33 | 44.9 | N/A |
| **T0pp** | 67 | 94.2 | 29 | 34.1 | 1.3 |
| *grd(·)*-NORMSAGEmini | 69 | 75.3 | 59 | 53.8 | 3.5 |
| *grd(·)*-NORMSAGE | 79 | 91.6 | **80** | **92.7** | 4.1 |

Table 8: Classification results (%) on norm grounding, along with rating (1–5) on the generated explanations.

onds per dialogue, and performs norm grounding at a rate of 3.8 seconds per dialogue. This process is done through OpenAI API, which may fluctuate slightly due to internet speed, but has the benefit of requiring no GPU usage from the researcher's end. The NORMSAGE norm discovery process is over 10x faster the human annotation efforts. The cost of OpenAI GPT-3 API access is currently $0.06 per 1k tokens, which is still less expensive than human annotation efforts. While quality may be a factor of difference, we showed that NORMSAGEmini, a variant of the proposed approach trained on silver-standard NORMSAGE outputs, perform competitively for norm discovery and verification tasks.

## 5   Related Work

The domain of norms is closely related to behavioral psychology and moral judgement. Early studies investigated the pragmatic cooperative principles (Grice, 1975), politeness implicatures (Kallia, 2004), and relationship between norms and law (Posner, 2009) governing human behavior. As

judgements of behavior are communicated through linguistics, (Graham et al., 2009) introduced a lexicon of evocative words based on moral foundation theory, which later attempts utilize for predicting the moral value from text messages (Lin et al., 2018; Mooijman et al., 2018). Recent approaches explore modeling moral and ethical judgement of real-life anecdotes from Reddit (Emelin et al., 2021; Sap et al., 2019a; Lourie et al., 2021; Botzer et al., 2022), with DELPHI (Jiang et al., 2021a) unifying the moral judgement prediction on these related benchmarks. Related is another line of work modeling legal judgement on judicial corpora (Chalkidis et al., 2022).

Norm discovery is a unique, emerging task, which aims to catalogue the underlying principles behind behavioral judgements, and can be seen as similar to distilling reactions, explanations, and implications from situations (Vu et al., 2014; Ding and Riloff, 2016; Rashkin et al., 2018; Sap et al., 2019b). Separate from explicit information extraction (Wen et al., 2021), such as the extraction of events/entities/relations which may tend to be overly granular and situation-specific, norm discovery involves inferring the underlying stand-alone abstracted social or cultural rules. Towards this end, Forbes et al. (2020); Ziems et al. (2022) are the main existing norm discovery approaches. Each presents a large-scale catalogue of manually curated rule-of-thumbs from Reddit post headers, and trains a language model to generate rule-of-thumbs based on this data. In contrast, our work focuses on norm discovery from conversations on-the-fly and without needing manual curation.

Modeling the social and moral dynamics in human interaction and communication have diverse applications, such as the detection of cyberbullying (Van Hee et al., 2015) and hate speech (Mathew et al., 2021), detoxification (Han et al., 2023), debiasing (Yu et al., 2023; Omrani et al., 2023; Yang et al., 2023), model alignment (Ouyang et al., 2022; Dong et al., 2023), bipartisan news framing (Fulgoni et al., 2016), social media understanding (Sun et al., 2023), emotions analysis (Zadeh et al., 2018; Yu et al., 2020), and situational report generation (Reddy et al., 2023), amongst other interdisciplinary human-centered NLP applications (Li et al., 2023a). In particular, discovering norms is essential for *explicitly* detecting norm adherence and violations instances (our work), as well as *implicitly* guiding dialogues (Ziems et al., 2022). From a

technical perspective, our norm discovery approach based on language model prompting and knowledge elicitation can be seen as a form of prompt engineering (Le Scao and Rush, 2021), where we prefix a question with an elaborated scene, with the underlying core intuition based on leveraging the zero-shot question answering capability of language models (Gangi Reddy et al., 2022). The norm grounding with explanation task is intuitively similar to the explainable natural language inference problem setting (Welleck et al., 2018; Wiegreffe et al., 2021). Our proposed framework, NORMSAGE, achieves norm discovery and grounding without intensive prompt-tuning (Jiang et al., 2021b) or finetuning (Forbes et al., 2020; Ziems et al., 2022).

## 6 Conclusions

We introduced a new NLP paradigm of guiding cross-culture communication with on-the-fly sociocultural-aware norm discovery, violation detection, and explanations. This conversation-based norm discovery and grounding problem goes beyond the US-centric data predominent in prior work. To address the new challenges, we present a framework called NORMSAGE that leverages knowledge elicitation from large language model (LM) prompting and incorporates novel self-verification mechanisms. Our approach surpasses baselines by improving the discovery of dialogue norms across diverse social and cultural groups, including multi-lingual conversations and culture-specific norms, while providing natural language explanations for interpretable norm discovery and violation detection. Our work has broad impacts as it empowers the discovery of norms which may differ from each other across cultures, thus enabling individuals to navigate communication barriers more effectively and find common grounds that avoid norm violations.

## Limitations

Our norm discovery process makes use of the GPT-3 from OpenAI[2] as a strong pre-trained language model to elicit groundable knowledge about the rules and judgements of acceptable behavior from human dialogue interactions. We recognize that norms may shift with context over time. Our discovery of norms applies to the time period that aligns with the conversation scenario in which a

---

[2] https://openai.com/api/

norm is discovered from. While the newer GPT-4 has now been released, we chose to stick with GPT-3 in our experiments due to the tight rate limit of GPT-4, currently capped at 25 request calls every 3 hours. We further point out that the GPT-3 model acquired its implicit knowledge from ultra large-scale data, and has added in mechanisms to address bias (Solaiman and Dennison, 2021). Nevertheless, all computational models still come with a risk of potential bias. For example, the behavior of some closed source models, such as ChatGPT, is not always guaranteed to be consistent over time (Chen et al., 2023). Moreover, explanations generated by language models may not always entail the models' predictions nor be factually grounded in the input (Ye and Durrett, 2022). We encourage researchers and practitioners to exercise caution and check-guards in their endeavors.

## Ethical Considerations

We recognize that the automatic generation of norms and judgements could be seen as normative and authoritative (Talat et al., 2021; Ziems et al., 2022). Here, we emphasize that the discovered norms are not intended to form a global and universally binding ethical system, but rather to provide a set of discrete intuitions and principles to help differentially explain the underlying assumptions that exist latently. The present work supports an explainable system to automatically discover norms from conversations on-the-fly, and verify whether the discovered norms can be sufficiently grounded to the data source, as well as the (entail vs. contradict) relation characteristic. Currently, we do not specifically distinguish between different levels in the cultural hierarchy (such as the relationship between Cantonese and Chinese culture), and resort to cultural references in the source data. It is also important to note that some automatically discovered norms might be viewed as stereotypical. Our transparent and flexible system should be seen as a human-AI collaboration tool that domain experts interface with, facilitating the moderation efforts.

**Privacy**   We abide by privacy guidelines for data used in our research study. For example, the SMS conversation data in the LDC source of Tab 1 has been collected through previous voluntary paid participation program such as BOLT (Song et al., 2014). Furthermore, the conversation data involved are utilized for research purpose only.

**Risks and Mitigations**   Our task involves source data that may contain explicit conversations about race, gender, religion, etc. We recognize the emotional burden that this presents to annotators (Roberts, 2016). In mitigation, we include the following content warning in the header of each task: *This HIT may contain text that disturbs some workers. If at any point you do not feel comfortable, please feel free to skip the HIT or take a break.* The study has been thoroughly reviewed and approved by a national level internal review board. The resources and findings presented in this work are intended for research purposes only. To ensure proper, rather than malicious, application of dual-use technology, we require users of our norm discovery data to complete a Data Usage Agreement that we link in our project repository. We also intend to make our software available as open source for public auditing, and explore measures to protect vulnerable groups.

## Acknowledgement

This research is based upon work supported by U.S. DARPA CCU Program No. HR001122C0034. The opinions, views and conclusions contained herein are those of the authors and should not be interpreted as necessarily representing the official policies, either expressed or implied, of DARPA or the U.S. Government. The U.S. Government is authorized to reproduce and distribute reprints for governmental purposes notwithstanding any copyright annotation therein.

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

# A Appendix

## A.1 Speaker Profiling and Cultural Indicator Extraction for Norm Discovery with Frames

As discussed in Sec 3, profiling speakers with cultural indicators help guide better structure and

culture-specific insights in norm discovery from conversations on-the-fly. The prompting procedure is intuitive with the presentation of the dialogue content and a directed question describing the task, similar to the prompting logic we introduced for conversation-based norm discovery, norm correctness verification, and relevance grounding. The specific prompt template used for cultural indicator extraction is:

> *Context:*
> $\{X_s\}$
>
> *For each person, extract the country- or state- level culture they are affiliated (in adjective form) if the information is available, and skip the person if the information is not available:*

For instance, the input for a dialogue snippet from "Outsourced" comes with a metadata description of summary from its Wikipedia page, containing text content such as:

> *"Todd Anderson (Josh Hamilton), a salesman for a Seattle novelty products company, learns he has to travel to India when his department is outsourced. Todd is not happy but when his boss Dave informs him that quitting would mean losing his stock options, he goes to train his Indian replacement Puro (Asif Basra)."*,

which we feed as value into $\{X_b\}$ in the prompt template. The output of this cultural indicator extraction is "Todd Anderson: American" and "Puro: Indian" in separate lines, which follows a consistent textual pattern that is easy to parse and is then plugged into the $dvr(\cdot)$-NORMSAGE input for norm discovery with framing.

## A.2  Norm Discovery Intrinsic Evaluation

### A.2.1  Assessment Metric Guidelines

We provide definitions of each intrinsic metric to the crowdsourced annotation workers.

- *Relevance*: is the norm inspired from the situation (lower bound on norm applicability).
- *Well-Formedness*: how well is the norm structured – is the norm self-contained, and does it include *both* a judgment of acceptability or occurrence, *and* an action or societal/cultural phenomena that is assessed.
- *Correctness*: to the best of their knowledge, do people agree that the described norm holds true?

- *Insightfulness*: does the norm convey enlightening understanding about what's considered acceptable and standard in the society that pertain to the conversation scenario.
- *Relatableness*: how well does the norm balance vagueness against specificity, so that it can generalize across multiple situations without being too specific.

We also provide good and bad examples of norms discovered, according to each of the metric dimensions. Consider a dialogue instance of *"I think we should divide the project tasks based on everyone's expertise and skills. It will ensure better efficiency and quality"*. We list good and bad examples of social norms with respect to the dialogue scenario as follows.

- Relevant social norm: "It's important to play to people's strengths and utilize their individual knowledge to deliver the best results."
  Irrelevant social norm: "It is good to earn money and retire early."
- Well-formed social norm: "It is good to utilize play to people's strength and deliver the best results".
  Poor-formed social norm: "Deliver the best results."
- Correct social norm: "Assigning tasks based on individual expertise and skills leads to better project outcomes".
  Incorrect social norm: "Assigning tasks based on the alphabetical order of employees' last names leads to better project outcomes".
- Insightful social norm: "Assigning tasks based on individual expertise not only maximizes efficiency and quality but also fosters a sense of ownership, motivation, and collaboration among team members."
  Non-insightful social norm: "It is nice to deliver good results."
- Relatable social norm: "Assigning tasks based on each team member's proficiency in relevant software tools (such as Excel for data analysis and presentation task) can lead to better project outcomes."
  Non-relatable social norm: "Assigning tasks based on each team member's proficiency in the specific Excel software tool leads to better project outcomes".

We ask annotators to rate social norms discovered from conversations on a 1-5 Likert scale, in which the spectrum of scoring guidelines is illustrated as:

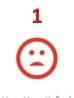 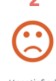 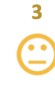 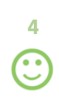 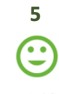