# OpenReview forum: "NORMSAGE: Multi-Lingual Multi-Cultural Norm Discovery from Conversations On-the-Fly"
_EMNLP/2023/Conference — EMNLP 2023 Main_

### Official Review · Reviewer_sQGj · 2023-08-01

**Soundness:** 3

**Excitement:**

4: Strong: This paper deepens the understanding of some phenomenon or lowers the barriers to an existing research direction.

**Missing References:**

For your limitations section: The Unreliability of Explanations in Few-shot Prompting for Textual Reasoning by Ye & Durrett NeurIPS 2022.

**Paper Topic And Main Contributions:**

The authors present a framework for automatically extracting social norms from English & Chinese conversation datasets. Their framework involves prompting GPT-3 to extract candidate norms, and then self-verify those norms. NORMSAGE outperforms other approaches at norm discovery, as determined by human evaluators in multiple experiments.

**Questions For The Authors:**

Line 250: How do you get the BERT embedding of a norm? (Embeddings are usually at the token-level rather than the phrase- or sentence-level). What is a BERT(n) embedding?

**Reasons To Accept:**

This paper is well-written and broadens research on social norms to additional perspectives beyond a predominantly white, U.S.-centric one. The experiments and findings are compelling and the results and examples are presented in a clear manner.

**Reasons To Reject:**

The authors use closed source models (e.g. GPT-3) that may be difficult to reproduce in future studies, as OpenAI models are often updated in unpredictable and non-transparent ways (e.g. “How is ChatGPT's behavior changing over time?” https://arxiv.org/abs/2307.09009).

A (small) limitation of this paper is that it assumes that a few select TV shows are sufficient to reflect entire cultures’ norms. I say “small” limitation because the author’s approach for extracting norms could be applied to additional data by future work that builds upon this current contribution.


**Reproducibility:**

3: Could reproduce the results with some difficulty. The settings of parameters are underspecified or subjectively determined; the training/evaluation data are not widely available.

**Reviewer Confidence:**

3: Pretty sure, but there's a chance I missed something. Although I have a good feel for this area in general, I did not carefully check the paper's details, e.g., the math, experimental design, or novelty.

**Typos Grammar Style And Presentation Improvements:**

Line 53: the citation is missing the year (2022).

Line 138-139: “better representing the diverse socioethnic or demographic groups” -> “which can better represent diverse socioethnic or demographic groups”

Line 178: “learnt” -> “learned”

Change “Muslim vs. Western” in Table 3 to “Pakistani vs. British” as you say “British” in Table 4. The authors do say they define culture based on “cultural references in the source data,” so this change would also assume a Pakistani vs. British comparison captures the dimension that is most emphasized by the writers of that particular show.

---

> ### Author Rebuttal · Authors · 2023-08-29
>
> We thank Reviewer sQGj for the positive review and thoughtful suggestion. We are pleased that the reviewer found our paper exciting in its approach to tackling an important research domain, supported by strong experimental results, compelling discussions/findings, and clear visualizations. We address the questions and feedback in the review as follows, and hope they clarify potential concerns.
>
> > **I.** Closed Source Model vs Open Source Model in the Framework Backbone
>
> We understand the reviewer’s concern that the current usage of the closed source GPT-3 model as backbone in our NormSage framework may involve a certain degree of limitation in reasoning transparency and predictability. We will enrich our paper with discussion about this, and include the relevant references you suggested. Moreover, with the recent advent of open-source language model alternatives that have also been finetuned to align with human feedback, such as Llama-2, we can add experimental results utilizing these open-sourced models as backbone in our NormSage framework for our camera-ready version, and we hope other research groups can further experiment with various models as NormSage’s backbone.
>
> > **II.** The current size of TV show selection as source of data input for cultural norm discovery
>
> The reviewer’s note that our norm discovery framework, NormSage, should be extended in future work to include a  broader selection of TV shows as source of input data to enable more sufficient reflection of an entire culture’s norm
> aligns indeed with our vision for the future directions built upon this work. For the present paper submission, 	we do acknowledge in the Limitations section (lines 621-626) that the discovered norms are not intended to form a global and universally binding ethical system, but rather to provide a set of discrete intuitions and principles to help differentially explain the underlying assumptions that exist latently in human-human interaction.
>
>
> ========
>
> Finally, in regards to how the BERT embedding for a norm is computed for similarity comparison, we derive the embedding pooled across the tokens in a text input through a *means()* operation. The current implementation in the transformer library for BERT models computes this pooled embedding, across the sentence-level, as one of its options. Our code is included, and the full software will be released to the rest of the NLP community as well upon paper publication.
>
> > from transformers import AutoTokenizer, AutoModel
> >
>
> > tokenizer = AutoTokenizer.from_pretrained("bert-base-uncased")
> >
> > model = AutoModel.from_pretrained("bert-base-uncased").eval()
> >
> >
> > ex_s = "It's important to communicate clearly and effectively."
> >
> > ex_s_toks = tokenizer(ex_s,return_tensors="pt")
> >
> > embed = model(**ex_s_toks).pooler_output
>
> > print(embed.size())   # embedding shape is (1, 768) signaling that mean pooling occurred across the sentence-level

---

### Official Review · Reviewer_HuvK · 2023-08-01

**Soundness:** 3

**Excitement:**

3: Ambivalent: It has merits (e.g., it reports state-of-the-art results, the idea is nice), but there are key weaknesses (e.g., it describes incremental work), and it can significantly benefit from another round of revision. However, I won't object to accepting it if my co-reviewers champion it.

**Paper Topic And Main Contributions:**

The paper introduces NORMSAGE to automatically extract culture-specific norms from multi-lingual conversations, though in reality it is bilingual conversations the authors addressed. They also provide relevance verification that can be extended to assess the adherence and violation of any norm with respect to a conversation on-the-fly, along with textual explanation.

**Reasons To Accept:**

The authors plan to release their data and code to the public, thereby aiding reproducibility.

The references seem adequate, relevant and recent enough.

Human annotators are involved in the evaluation of the work and some details of the process of hiring them are discussed.

**Reasons To Reject:**

Although the paper mentions existing gaps in the field it covers and its contributions, it does not explicitly state its research question or objectives, making it hard to determine if the paper’s results answer its question.

The reason for the choices of the threshold of 0.7 and 0.6 for the correctness probability and grounding score, respectively, are not given.

Line 256 makes a claim about no prior work without any reference or mentioning what study the authors carried out that informed this claim.

The structure of the paper seems a bit scattered – where there are several Result paragraphs under different sections, resulting in tables of results being mixed with tables of methods. Also, line 256 to 310, which one would expect under a Methodology section, is under Evaluation and Results.

Line 382 makes a statement about norm correctness without any reference or justification.

Some of the examples provided (e.g. Table 6) have grammar/correctness issues and it is unclear if this may have affected the results as presented.

**Reproducibility:**

3: Could reproduce the results with some difficulty. The settings of parameters are underspecified or subjectively determined; the training/evaluation data are not widely available.

**Reviewer Confidence:**

4: Quite sure. I tried to check the important points carefully. It's unlikely, though conceivable, that I missed something that should affect my ratings.

**Typos Grammar Style And Presentation Improvements:**

The work focused on two languages (bilingual) instead of multilingual, as claimed in the paper.

Grammar check and proof-reading required. For example line 36 “cability”, line 357, Fleiss not “Fless”.

Some of the in-text references are not adequately formatted, e.g. line 53.

---

> ### Author Rebuttal · Authors · 2023-08-29
>
> We thank Reviewer HuvK for the review. We are glad that the reviewer found our paper well-motivated in relation to the diverse related work in the field, and well-documented in experimental details (ranging from method implementation and data setting for reproducibility, to the human evaluation process). In response to questions and feedback raised in the review, we address them as follows and hope our response clarify possible misconceptions or concerns you may have.
>
> > **I.** Our Research Questions/Objectives and whether Results Address Them
>
> Our work focuses on addressing the core research questions of i) how can norms insightful for guiding human-human interaction be effectively extracted from diverse conversational sources across different cultures and in different languages; and ii) can we validate the norms discovered within conversational contexts in order to improve robustness against hallucination in language model generation, while also considering cultural and social nuances.
>
> We address these research questions through NormSage, our proposed framework for norm discovery with self-verification. Specifically, with regards to the first research question on extrinsic norm discovery, we show in Sec 4.1 and Table 2 that our method discovers significantly more relevant and insightful norms for conversations on-the-fly (selected as "best" ≥70% cases) compared to baselines such as NMT and pMT developed over Reddit-based rules-of-thumb annotations. Additionally, the norms discovered from Chinese conversation exhibit limited performance differences compared to the norms discovered from English conversation in terms of relevance and correctness (∆ ≤ 5%), as presented in lines 332-334 of Sec 4.1 results. We further show in Sec 4.1.1 that the culture-specific norms are also promising in quality, with humans being able to identify the culture from these norms at 80% accuracy.
>
> With regards to the second research question on norm reliability verification, we demonstrate through correctness prediction experiments in Sec 4.2 and grounding prediction experiments in Sec 4.3 the superiority of NormSage over pre-existing baselines. In particular, NormSage achieves an AUC of 94.6% in three-class classification for grounding, with generated explanations matching human-written quality.
>
> > **II.** Threshold Selection for the filtering process of NormSage in Correctness Verification and Grounding Classification
>
> We randomly selected a small sample of ten data points that are held out separately from the test set used in Section 4.2-4.3 evaluation, and searched at increments of 0.1 for the threshold value that is optimal for correctness and grounding filtering. Our small-scale threshold comparison results will be further detailed into our paper appendix, and the exact data sample used will be made public for reproducibility.
>
> > **III.** Claim about no Prior Work in Line 256
>
> We are not aware of any prior work on norm discovery from multilingual conversations across cultures. We positioned our work with respect to other work on computational social norms in related work. We would welcome any references that rebut our claim.
>
> > **IV.** Presentation Structure of our Paper
>
> Regarding our paper’s structure, we clarify that the sole table/figure under the Methodology section (i.e., Fig 2) is to present a high-level motivational overview of our two versions of prompt templates for initial norm discovery, with and without framing structure, which we believed were a crucial component to understanding how the proposed dvr() mechanism in our NormSage framework operates. In contrast, the tables of visualizations under Experiments and Results (i.e., Tables 4,6) are designed to present more comprehensive coverage and details on the results of the primary norm discovery task and secondary verification tasks, which were not immediately crucial to the understanding of the core method proposed.
>
> In particular, Table 4 under Experiment and Results is intended to illustrate various norms covering different topics that are discovered from different cultures, while further highlighting norms that contrast each other across certain cultures. Table 6 under Experiment and Results is intended to illustrate, through detailed examples, results grounding norms to dialogue situations, for each of the grounding classes <contradict, irrelevant, entail>, with utterance-level provenance underlined. Because these details are insightful for results analysis but not immediately crucial for understanding the grounding mechanism, which can be sufficiently described in detail by line 232-240 in Methods, we believe it is most reasonable to include this information under Experiment and Results as opposed to the Methodology section.
>
> Finally, lines 256-310 is a description of our choice of baseline methods for experimental comparison, which we believe is fitting to be included under the Experiment and Results, as the majority of other works do in EMNLP/ACL-related venues.
>
> We hope this explanation helps address your previous concern and clear potential misunderstanding for this aspect here.
>
> > **V.** Clarification on the line 382 statement of our norm correctness verification process
>
> In Line 382, we wrote that “norm correctness verification is a two-class classification problem”. This statement falls in line with how we defined our norm correctness verification task in Sec 2.1, as a classification problem involving two classes with a norm being either correct or incorrect. Our intention in line 382 is to refresh the reader’s memory of the task when we lay out the evaluation metrics applicable to the task in the Sec 4.2 experiments on norm correctness verification.
>
> > **VI.** Clarification about Potential Grammar/Correctness Issues in Table 6 and the effect on Results
>
> We acknowledge that the dialogue inputs shown in the experimental results visualization of Table 6 may contain improper grammar. However, we point out that this is exactly the nature of real-world conversations, involving noisy or informally styled textual dialogue, which makes our novel norm discovery on-the-fly task setting introduced unique and challenging. We will explicitly clarify this in our paper. Moreover, despite the noisy nature of dialogue inputs, Table 6 shows that our large language model prompting mechanism can still effectively reason over the <entailment, irrelevance, contradiction> nature of grounding between a given dialogue input and a given norm with proper explanations.

---

### Official Review · Reviewer_6CHs · 2023-08-03

**Typos Grammar Style And Presentation Improvements:** N/A
**Soundness:** 3

**Excitement:**

3: Ambivalent: It has merits (e.g., it reports state-of-the-art results, the idea is nice), but there are key weaknesses (e.g., it describes incremental work), and it can significantly benefit from another round of revision. However, I won't object to accepting it if my co-reviewers champion it.

**Missing References:**

N/A

**Paper Topic And Main Contributions:**

This paper proposed a new framework, NormSage, to automatically extract culture-specific norms from multi-lingual conversations. Specifically, NormSage uses GPT-3 prompting to extract norms directly from conversations and provides explainable self-verification to ensure correctness and relevance.

**Questions For The Authors:**

1. What is your positioning for your articles? A tech paper, resource paper or position paper?

**Reasons To Accept:**

1. A promising track. Automatic norm discovery in the multi-lingual multi-cultural context is useful for those cross-cultural conversations.

2. A large-scale Norm knowledge base has been proposed for further exploration.

3. Overall writing is in good format and easy to follow. Detailed baseline code and data samples are provided for robustness demonstration.

**Reasons To Reject:**

1. My main concern is about the contributions: (1) Although this paper claimed their first trial on multilingual and multi-cultural norm discovery in multi-turn conversations, some previous multi-lingual information extraction work and tools can be easily applied here. Besides, I cannot find any specific design in NormSage to resolve the multi-cultural challenges (Please correct me if I'm wrong). It seems that this framework can be applied to other tasks? (2) NormSage is a reasonable pipeline framework to prompt GPT-3 to automatically discover and verify norms. Nevertheless, the contribution of a framework with several simple prompts is not sufficient for a technical paper.

2. NormSage is a GPT-3 based framework; however, most baselines in Table 2 are much smaller than GPT-3, such as GPT-2. That's not enough to demonstrate the advantages of NormSage. More competitive baseline models should be incorporated. The same problem occurs for Table 7 regarding the comparison between NormSage (GPT-3 based) and BART or T5-based models.

3. Lack of illustration about how to use the NormKB for downstream tasks.

**Reproducibility:**

4: Could mostly reproduce the results, but there may be some variation because of sample variance or minor variations in their interpretation of the protocol or method.

**Reviewer Confidence:**

5: Positive that my evaluation is correct. I read the paper very carefully and I am very familiar with related work.

---

> ### Author Rebuttal · Authors · 2023-08-29
>
> We thank Reviewer 6CHs for the review. We appreciate the reviewer's recognition of our paper in tackling a useful/promising task domain, and also its well-writteness with detailed examples and resources for future work to explore further upon. We address the questions and feedback raised in the review as follows, and hope they clarify potential misconceptions or concerns of our work.
>
> > **I.** On the applicability of previous multilingual information extraction (IE) work and tools for norm discovery
>
> Norms are often implicitly mentioned, so they need to be inferred from the entire dialogue discourse. Most IE work, on the other hand, focuses on explicitly extracting specific pieces of information (e.g., entities, events, relations, etc.) that are often too granular and situation-specific to represent a stand-alone abstracted social or cultural norm. In reference to the Figure 1 input dialogue as example, IE may parse out entities such as “I” (the speaker) and “xiaolongbao”, and events such as “threw”. Yet, the norm discovery task aims to identify the underlying social or cultural rules, such as “It is good to be honest with your family”, which are often not readily apparent from the results of explicit information extraction. Due to this reason, we did not include information extraction work into our current set of baselines for norm discovery. But if there is a particular IE tool that you strongly recommend, we can look into incorporating it for our camera-ready version.
>
> > **II.** Clarification on how NormSAGE helps solve multi-cultural challenges
>
> We motivate in the Introduction and Task Formulation sections that knowledge of norms in general, and sociocultural norms in particular, is essential if we are to equip AI systems with capability to understand and reason about acceptable behavior in communication and interaction across cultures. We concretize this objective in the Experiment and Results section through visualization of culture-specific norms discovered, and further highlight the norms that contrast each other across specific cultures (please refer to Table 6). For example, in Pakistani culture it is not uncommon for the bride and groom to not meet each other until the wedding day, whereas in British culture, it is normal for the bride and groom to meet each other before the wedding. Heated events revolving around this cultural clash actually occur in “Citizen Khan” (a TV show about a British Pakistani family), and we extract this contrasting pair of norms from the source conversation data. An important impact of our work is that discovering such norms can help humans better navigate communication challenges and find common grounds that avoid norm violations. We demonstrate this key concept through effective empirical performances of grounding (i.e., identifying the <entail,contradict,irrelevant> relation between a given norm and a dialogue input) in Sec 4.3. We will make these points clearer with the extra page allotment for the camera-ready version.
>
> > **III.** Framework Novelty and Contribution
>
> The technical contribution of our NormSage framework lies in the fundamental design philosophy of leveraging correctness and relevance verification for enabling norm discovery on-the-fly and addressing the important LM hallucination problem. We empirically show that our two-step verification mechanism proposed in NormSage improves the intrinsic property of discovered norms by over 14% in relevance and correctness (on an absolute scale), through simple yet effectively designed prompting mechanisms. Our study findings reveal important insights that large language models are capable of performing knowledge elicitation and self-verification to effectively reason over conversation scenarios across cultures and languages, discover underlying norms insightful for facilitating harmonious human interactions, and also detect norm violations with grounded explanations.
>
> > **IV.** Choice of Baselines and Model Size
>
> We would like to address the concern about competitive baselines. Our current choice of baseline models were selected based on either domain relevance (e.g., LM variants trained or based on SocialChem and Moral Integrity Corpus) or their established/published zero-shot performance across NLP tasks (e.g., the T5-based T0pp), while bearing in mind to keep methods that are runnable and accessible on a typical CPU machine or GPU that is V100 or less in size. It is thus true that the baselines in Table 2, such as the pMT model adapted from GPT-2, are smaller than GPT-3 in parameter size. However, we clarify that the fundamental design of our NormSage framework in combining initial norm discovery with correctness and grounding self-verification mechanisms can be extended beyond its current GPT-3 backbone to other powerful large language model backbones, such as LLaMA-2 70B, as well as compact language models finetuned on prediction data from NormSage (i.e., the NormSage_mini method in our paper).
>
> > **V.** Applying NormKB for downstream tasks
>
> We agree that we can improve the submission through additional illustration of how to use NormKB for downstream tasks. However, we believe that our current experimental setup and findings in the primary task of norm discovery and the subsidiary tasks of norm correctness and grounding verification, which construct the NormKB, deserve to be considered already as important milestones for knowledge acquisition in socially and culturally important norms that guide human-human interactions. We also point out that the grounding of norms to a dialogue input to identify entailment/contradiction/irrelevance relationship (Sec 4.3) extends to identifying the adherence or violation occurrence of any norm in the NormsKB to a particular conversation. We plan to elaborate this further with clearer examples and explanations in the next version of our paper. For future research directions, we believe it may be meaningful to explore leveraging NormsKB as a data resource for pretraining large language models to be better equipped with commonsense knowledge that aligns with human preferences.
>
> ========
>
> > Question regarding the positioning of our paper
>
> Overall, our submission is designed as a research paper with primary contributions in: i)  presenting an important new task of conversation-based multilingual and multicultural norm discovery, which is useful for guiding harmonious human communication interactions; and ii)  proposing a technical approach for tackling challenges involved in the task (i.e., the prompt-based framework, NormSage, with novel self-verification mechanisms and designs supporting explainability).  Our technical contributions also include the setup of comprehensive evaluation objectives (e.g., norm correctness, pairwise culture comparison identification, etc.) to benchmark language model performance for the new task. We point out that these involve insightful discussions, such as the difference between 2-class and 3-class grounding classification in terms of design considerations and model performance observations (lines 446-455, 486-492). Beyond technical contributions, we also present NormsKB as a useful resource for future research work, thought this aspect is positioned as more of a ramification impact from our work worthwhile to point out.

---

### Meta-Review · Area_Chair_3JCh · 2023-09-23

**Recommendation:** 4

**Metareview:**

This is a very interesting paper that proposes an approach to automatically extract cultural norms from multilingual conversations. A strong reason to accept this is that this paper presents an effective and innovative solution to an under-explored, important problem which will broaden the NLP field by looking beyond the cultural norms of English-spoken western societies.

While there were concerns about the novelty of the work, pointing out that previous work in IE can be applied here, authors addressed that concern in a convincing way that social norms cannot be described in a word or phrase which is what IE does. The concerns about the language models used and the narrow scope of the data are valid, but I think the authors' replies are sufficient. I would like to see additional experiments done with open-source large language models, as well as discussions about extracting social norms from other sources.

Overall, this is fascinating work which will lead to further research in this important topic.

---

### Decision · Program_Chairs · 2023-10-07

**Decision:**

Accept-Main

**Comment:**

This is a very interesting paper that proposes an approach to automatically extract cultural norms from multilingual conversations. A strong reason to accept this is that this paper presents an effective and innovative solution to an under-explored, important problem which will broaden the NLP field by looking beyond the cultural norms of English-spoken western societies.

While there were concerns about the novelty of the work, pointing out that previous work in IE can be applied here, authors addressed that concern in a convincing way that social norms cannot be described in a word or phrase which is what IE does. The concerns about the language models used and the narrow scope of the data are valid, but I think the authors' replies are sufficient. I would like to see additional experiments done with open-source large language models, as well as discussions about extracting social norms from other sources.

Overall, this is fascinating work which will lead to further research in this important topic.